# DIFFERENTIALLY PRIVATE MIXED-TYPE DATA GENERATION FOR UNSUPERVISED LEARNING

## ABSTRACT

In this work we introduce the DP-auto-GAN framework for synthetic data generation, which combines the low dimensional representation of autoencoders with the flexibility of Generative Adversarial Networks (GANs). This framework can be used to take in raw sensitive data, and privately train a model for generating synthetic data that will satisfy the same statistical properties as the original data. This learned model can be used to generate arbitrary amounts of publicly available synthetic data, which can then be freely shared due to the post-processing guarantees of differential privacy. Our framework is applicable to unlabeled *mixed-type data*, that may include binary, categorical, and real-valued data. We implement this framework on both unlabeled binary data (MIMIC-III) and unlabeled mixed-type data (ADULT). We also introduce new metrics for evaluating the quality of synthetic mixed-type data, particularly in unsupervised settings.

## 1 INTRODUCTION

As data storage and analysis are becoming more cost effective, and data become more complex and unstructured, there is a growing need for sharing large datasets for research and learning purposes. This is in stark contrast to the previous statistical model where a data curator would hold datasets and answer queries from (potentially external) analysts. Sharing entire datasets allows analysts the freedom to perform their analyses in-house with their own devices and toolkits, without having to pre-specify the analyses they wish to perform. However, datasets are often proprietary or sensitive, and cannot be shared directly. This motivates the need for *synthetic data generation*, where a new dataset is created that shares the same statistical properties as the original data. These data may not be of a single type: all binary, all categorial, or all real-valued; instead they may be of *mixed-types*, containing data of multiple types in a single dataset. These data may also be unlabeled, requiring techniques for *unsupervised learning*, which is typically a more challenging task than supervised learning on labeled data.

Privacy challenges naturally arise when sharing highly sensitive datasets about individuals. Ad hoc anonymization techniques have repeatedly led to severe privacy violations when sharing "anonymized" datasets. Notable examples include the Netflix Challenge (Narayanan & Shmatikov, 2008), the AOL Search Logs (Barbaro & Zeller, 2006), and Massachusetts State Health data (Ohm, 2010), where linkage attacks to publicly available auxiliary datasets were used to reidentify individuals in the dataset. Even deep learning model have been shown to inadvertently memoize sensitive personal information such as Social Security Numbers during training (Carlini et al., 2019).

Differential privacy (DP) (Dwork et al., 2006) (formally defined in Section 2) has become the de facto gold standard of privacy in the computer science literature. Informally, it bounds the amount the extent to which an algorithm can depend on a single datapoint in its training set. This guarantee ensures that any differentially privately learned models do not overfit to individuals in the database, and therefore cannot reveal sensitive information about individuals. It is an information theoretic notion that does not rely on any assumptions of an adversary's computational power or auxiliary knowledge. Furthermore, it has been shown empirically that training machine learning models with differential privacy protects against membership inference and model inversion attacks (Triastcyn & Faltings, 2018; Carlini et al., 2019). Differentially private algorithms have been deployed at large scale in practice by organizations such as Apple, Google, Microsoft, Uber, and the U.S. Census Bureau.

Much of the prior work on differentially private synthetic data generation has been either theoretical algorithms for highly structured classes of queries (Blum et al., 2008; Hardt & Rothblum, 2010) or based on deep generative models such as Generative Adversarial Networks (GANs) or autoencoders. These architectures have been primarily designed for either all-binary or all-real-valued datasets, and have focused on the *supervised* setting, where datapoints are labelled.

In this work we introduce the *DP-auto-GAN framework*, which combines the low dimensional representation of autoencoders with the flexibility of GANs. This framework can be used to take in raw sensitive data, and privately train a model for generating synthetic data that should satisfy the same statistical properties as the original data. This learned model can be used to generate arbitrary amounts of publicly available synthetic data, which can then be freely shared due to the post-processing guarantees of differential privacy. We implement this framework on both unlabeled binary data (for comparison with previous work) and unlabeled mixed-type data. We also introduce new metrics for evaluating the quality of synthetic mixed-type data, particularly in unsupervised settings.

## 1.1 OUR CONTRIBUTIONS

In this work, we provide three main contributions: a new algorithmic framework for privately generating synthetic data, new evaluation metrics for measuring the quality of synthetic data in unsupervised settings, and empirical evaluations of our algorithmic framework using our new metrics, as well as standard metrics.

**Algorithmic Framework.** We propose a new data generation architecture which combines the versatility of an autoencoder (Kingma & Welling, 2013) with the recent success of GANs on complex data. Our model extends previous autoencoder-based DP data generation (Abay et al., 2018; Chen et al., 2018) by removing an assumption that the distribution of the latent space follows a mixture of Gaussian distribution. Instead, we incorporate GANs into the autoencoder framework so that the generator must learn the true latent distribution against the discriminator. We describe the composition analysis of differential privacy when the training consists of optimizing both autoencoders and GANs (with different noise parameters). Furthermore, in this analysis we halve the noise injected into autoencoder from all existing works while provably maintaining the same mathematical privacy guarantee.

**Unsupervised-Learning Evaluation Metric of Synthetic Data.** We define several new metrics that evaluate the performance of synthetic data compared to the original data when the data is of *mixed-type*. Previous metrics in the literature are applicable only to all-binary or all-real-valued datasets. Our new metrics generalize the previously used metrics (Choi et al., 2017; Xie et al., 2018) from all-binary data to mixed-type by training various learning models to predict each feature from the rest of the data in order to assess correlation between features. In additional, our metrics do not require a particular feature to be specified as a label, and therefore do not assume a supervised-learning nature of the data, as in much of the previous work does (Papernot et al., 2017; 2018; Jordon et al., 2018).

**Empirical Results.** We empirically comepare the performance of our algorithmic framework on the MIMIC-III medical dataset (Johnson et al., 2016) and UCI ADULT Census dataset (Dua & Graff, 2017) using previously studied metrics in literature (Frigerio et al., 2019; Xie et al., 2018). Our experiments show that our algorithms perform better, and allow significantly improved $\epsilon$ values with $\epsilon \approx 1$, compared to prior work (Xie et al., 2018) with $\epsilon \approx 200$. We evaluate our synthetic data using new quantitative and qualitative metrics, confirming that the performance of our algorithm remains high even for small values of $\epsilon$, corresponding to strong privacy guarantees. Our code is made publicly available for future use and research.

## 1.2 RELATED WORK ON DIFFERENTIALLY PRIVATE DATA GENERATION

Early work on differentially private synthetic data generation was focused primarily on theoretical algorithms for solving the *query release problem* of privately and accurately answering a large class of pre-specified queries on a given database. It was discovered that generating synthetic data on which the queries could be evaluated allowed for better privacy composition than simply answering all the queries directly (Blum et al., 2008; Hardt & Rothblum, 2010; Hardt et al., 2012; Gaboardi et al., 2014). Bayesian inference has also been used for differentially private data generation (Zhang

et al., 2017; Ping et al., 2017) by estimating the correlation between features. See Surendra & Mohan (2017) for a survey of techniques used in private synthetic data generation through 2016.

In 2016, Abadi et al. (2016) introduced a framework for training deep learning models with differential privacy. Non-convex optimization, which is required when training deep models, can be made differentially private by adding a Gaussian noise to a clipped (norm-bounded) gradient in each training step. Abadi et al. (2016) also introduced the *moment accountant* privacy analysis for private stochastic gradient descent, which provided much tighter Gaussian-based privacy composition and allowed for significant improvements in accuracy over previously used composition techniques, such as advanced composition Dwork et al. (2010). The moment account was later defined in terms of *Renyi Differential Privacy (RDP)* (Mironov, 2017), which is a slight variant of differential privacy designed for easy composition, particularly for differentially private stochastic gradient descent (DP-SGD). Much of the work that followed on private data generation used deep (neural-network-based) generative models to generate synthetic data, and can be broadly categorized into two types: autoencoder-based and GAN-based. Our algorithmic framework is the first to combine both DP GANs and autoencoders into one framework.

**Differentially Private Autoencoder-Based Models.** A variational autoencoder (VaE) (Kingma & Welling, 2013) is a generative model that compresses high-dimensional data to a smaller space called *latent space*. The compression is commonly achieved through deep models and can be differentially privately trained (Chen et al., 2018; Acs et al., 2018). VaE makes the (often unrealistic) assumption that the *latent distribution* is Gaussian. Acs et al. (2018) uses Restricted Boltzmann machine (RBM) to learn the latent Gaussian distribution, and Abay et al. (2018) uses expectation maximization to learn a Gaussian mixture. Our work extends this line of work by additionally incorporating the generative model GANs which have also been shown to be successful in learning latent distributions.

**Differentially Private GANs.** GANs are a generative model proposed by Goodfellow et al. (2014) that have been shown success in generating several different types of data (Mogren, 2016; Saito et al., 2017; Salimans et al., 2016; Jang et al., 2016; Kusner & Hernández-Lobato, 2016; Wang et al., 2018). As with other deep models, GANs can be trained privately using the aforementioned private stochastic gradient descent (formally introduced in Section 2.1). Additional related work, including variants of the DP GAN framework, optimization techniques to improve the performance of DP GANs, and Table 4 summarizing these works can be found in Appendix D.

**Differentially Private Generation of Mixed-Type Data.** Next we describe the three most relevant recent works on privately generating synthetic data of mixed type. Abay et al. (2018) consider the problem of generating mixed-type labeled data with $k$ possible labels. Their algorithm, DP-SYN, partitions the dataset into $k$ sets based on the labels and trains a DP autoencoder on each partition. Then a DP expectation maximization (DP-EM) algorithm of Park et al. (2017) is used to learn the distribution in the latent space of encoded data of the given label-class. The main workhorse, DM-EM algorithm, is designed and analyzed for Gaussian mixture models and more general factor analysis models. Chen et al. (2018) works in the same setting, but replaces the DP auto-encoder and DP-EM with a DP variational auto-encoders (DP-VaE). Their algorithm assumes that the mapping from real data to the Gaussian distribution can be efficiently learned by the encoder. Finally, Frigerio et al. (2019) used a Wasserstein GAN (WGAN) to generate differentially private mixed-type synthetic data. This type of GAN uses a Wasserstein-distance-based loss function in training. Their algorithmic framework privatized the WGAN using DP-SGD, similar to the previous approaches for image datasets (Zhang et al., 2018; Xie et al., 2018). The methodology of Frigerio et al. (2019) for generating mixed-type synthetic data involved two main ingredients: changing discrete (categorical) data to binary data using one-hot encoding, and adding an output softmax layer to the WGAN generator for every discrete variable.

Our framework is distinct from these three approaches. We use a differentially private auto-encoder which, unlike DP-VaE of Chen et al. (2018), does not require mapping data to a Gaussian distribution. This allows us to reduce the dimension of the problem handled by the WGAN, hence escaping the issues of high-dimensionality from the one-hot encoding of Frigerio et al. (2019). We also use DP-GAN, replacing DP-EM in Abay et al. (2018), for learning distributions in the latent encoded space.

**Evaluation Metrics for Synthetic Data.** Various evaluation metrics have been considered in the literature to quantify the quality of the synthetic data (see Charest (2011) for a survey). The metrics

can be broadly categorized into two groups: *supervised* and *unsupervised*. Supervised evaluation metrics are used when there are clear distinctions between features and labels of the dataset, e.g., for healthcare applications, a person's disease status is a natural label. In these settings, a predictive model is typically trained on the synthetic data, and its accuracy is measured with respect to the real (test) dataset. Unsupervised evaluation metrics are used when no feature of the data can be decisively termed as a label. Recently proposed metrics include *dimension-wise probability* for binary data (Choi et al., 2017), which compares the marginal distribution of real and synthetic data on each individual feature, and *dimension-wise prediction* which measures how closely synthetic data captures relationships between features in the real data. This metric was proposed for binary data, and we extend it here to mixed-type data. Recently, NIST (2019) used a 3-way marginal evaluation metric which used three random features of the real and synthetic datasets to compute the total variation distance as a statistical score. See Appendix D for more details on both categories of metrics, including Table 1 which summarizes the metrics' applicability to various data types.

## 2 PRELIMINARIES ON DIFFERENTIAL PRIVACY

In the setting of differential privacy, a dataset $X$ consists of $m$ individuals' sensitive information, and two datasets are neighbors if one can be obtained from the other by the addition or deletion of one datapoint. Differential privacy requires that an algorithm produce similar outputs on neighboring datasets, thus ensuring that the output does not overfit to its input dataset, and that the algorithm learns from the population but not from the individuals.

**Definition 1** (Differential privacy (Dwork et al., 2006)). *For $\epsilon, \delta > 0$, an algorithm $\mathcal{M}$ is $(\epsilon, \delta)$-differentially private if for any pair of neighboring databases $X, X'$ and any subset $S \subseteq Range(\mathcal{M})$,*

$$\Pr[\mathcal{M}(X) \in S] \le e^\epsilon \cdot \Pr[\mathcal{M}(X') \in S] + \delta.$$

A smaller value of $\epsilon$ implies stronger privacy guarantees (as the constraint above binds more tightly), but usually corresponds with decreased accuracy, relative to non-private algorithms or the same algorithm run with a larger value of $\epsilon$. Differential privacy is typically achieved by adding random noise that scales with the *sensitivity* of the computation being performed, which is the maximum change in the output value that can be caused by changing a single entry. Differential privacy has strong *composition guarantees*, meaning that the privacy parameters degrade gracefully as additional algorithms are run on the same dataset. It also has a *post-processing* guarantee, meaning that any function of a differentially private output will maintain the same privacy guarantees.

### 2.1 DIFFERENTIALLY PRIVATE STOCHASTIC GRADIENT DESCENT (DP-SGD)

The DP-SGD framework (given formally in Algorithm 5 in Appendix D.1) is generically applicable for private non-convex optimization. In our proposed model, we use this framework to train the autoencoder and GAN.

Training deep learning models reduces to minimizing some (empirical) loss function $f(X;\theta) := \frac{1}{m}\sum_{i=1}^{m} f(x_i;\theta)$ on a dataset $X = \{x_i \in \mathbb{R}^n\}_{i=1}^{m}$. Typically $f$ is a nonconvex function, and a common method to minimize $f$ is by iteratively performing stochastic gradient descent (SGD). To make SGD private, Abadi et al. (2016) proposed to is to first clip the gradient of each sample to ensure bounded $\ell_2$-norm, and then add multivariate Gaussian noise to the gradient. The clipping reduces the scale of noise that must be added to preserve differential privacy. The noisy-clipped-gradient is then used in the update step instead of the true gradient. Further details of this procedure are deferred to Appendix D.1.

A variant notion of differential privacy, known as *Renyi Differential Privacy (RDP)* (Mironov, 2017), that is often used to analyze privacy for DP-SGD. A randomized mechanism $\mathcal{M}$ is $(\alpha, \epsilon)$-RDP if for all neighboring databases $X, X'$ that differ in at most one entry,

$$RDP(\alpha) := D_\alpha(\mathcal{M}(X)||\mathcal{M}(X')) \le \epsilon,$$

where $D_\alpha(P||Q) := \frac{1}{\alpha-1} \log \mathbb{E}_{x \sim X}\left(\frac{P(x)}{Q(x)}\right)^\alpha$ is the *Renyi divergence* or *Renyi entropy* of order $\alpha$ between two distributions $P$ and $Q$. Renyi divergence is better tailored to tightly capture the privacy loss from the Gaussian mechanism that is used in DG-SGD, and is a common analysis tool

for DP-SGD literature. To compute the final $(\epsilon, \delta)$-differential privacy parameters from iterative runs of DP-SGD, one must first compute the subsampled Renyi Divergence, then compose privacy under RDP, and then convert the RDP guarantee into DP. Further details of this process are given in Appendix D.2.

## 3 ALGORITHMIC FRAMEWORK

The overview of our algorithmic framework DP-auto-GAN is shown in Figure 1, and the full details are given in Algorithm 1. The algorithm takes in $m$ raw data points, and *pre-processes* these points into $m$ vectors $x_1, \ldots, x_m \in \mathbb{R}^n$ to be read by DP-auto-GAN, where usually $n$ is very large. For example, categorical data may be pre-processed using one-hot encoding, or text may be converted into numerical values. Similarly, the output of DP-auto-GAN can be *post-processed* from $\mathbb{R}^n$ back to the data's original form. We assume that this pre- and post-processing can done based on public knowledge, such as possible categories for qualitative features and reasonable bounds on quantitative features, and therefore does not require privacy.

Within the DP-auto-GAN, there are two main components: the *autoencoder* and the GAN. The autoencoder serves to reduce the dimensionality of the data before it is fed into the GAN. The GAN consists of a *generator* that takes in noise $z$ sampled from distribution $Z$ and produces $G_w(z) \in \mathbb{R}^d$, and a *discriminator* $D_y(\cdot) : \mathbb{R}^n \to \{0, 1\}$. Because of the autoencoder, the generator only needs to synthesize data based on the latent distribution $\mathbb{R}^d$, which is a much easier task than synthesizing in the original high-dimensional space $\mathbb{R}^n$. Both components of our architecture, as well as our algorithm's overall privacy guarantee, are described in the remainder of this section.

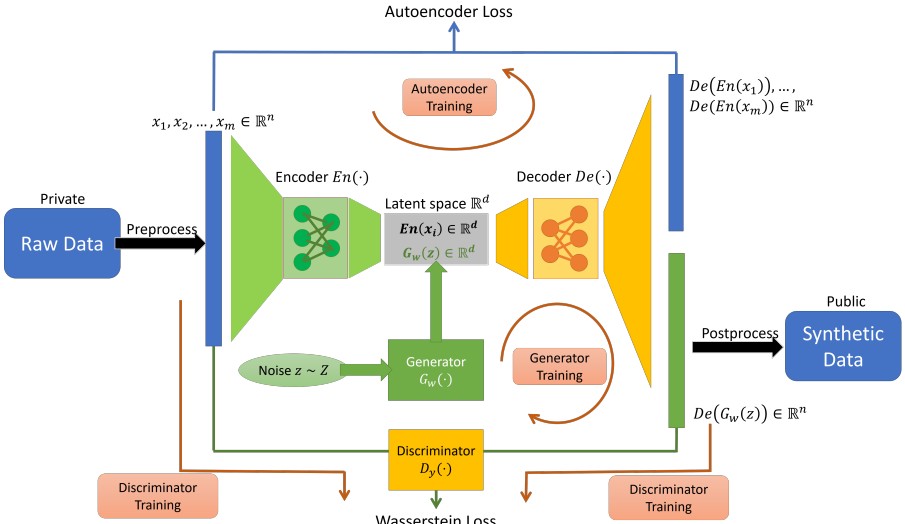

Figure 1: The summary of our DP-auto-GAN algorithmic framework. Pre- and post-processing (in black) are assumed to be public knowledge. Encoder and generator (in green) are trained without noise injection, whereas decoder and discriminator (in yellow) are trained with noise. The four red arrows indicate how data are forwarded for each training: autoencoder training, generator training, and discriminator training. After training, the generator and decoder (but not encoder) are released to the public to generate synthetic data.

### 3.1 AUTOENCODER TRAINING

The autoencoder consists of the encoder $En_\phi(\cdot) : \mathbb{R}^n \to \mathbb{R}^d$ and decoder $De_\theta(\cdot) : \mathbb{R}^d \to \mathbb{R}^n$ parametrized by edge weights $\phi, \theta$, respectively. The architecture of the autoencoder assumes that high-dimensional data $x_i \in \mathbb{R}^n$ can be represented compactly in low-dimensional space $\mathbb{R}^d$, also called *latent space*. The encoder $En_\phi$ is trained to find such low-dimensional representations. We also need the decoder, $De_\theta$, to map this point $En_\phi(x_i)$ in the latent space back to $x_i$. A measure of

the information preserved in this process is the error between the decoder's image and the original $x_i$. Thus a good autoencoder should minimize the distance $\text{dist}(De_\theta(En_\phi(x_i)), x_i)$ for each data-point $x_i$ and the appropriate distance function dist. Our autoencoder uses binary cross entropy loss: $\text{dist}(x, y) = -\sum_{j=1}^n y_{(j)} \log(x_{(j)}) - \sum_{j=1}^n (1 - y_{(j)}) \log(1 - x_{(j)})$ (where $x_{(j)}$ is the $j$th coordinate of $x \in \mathbb{R}^n$).

This also motivates the definition of a (true) loss function $\mathbb{E}_{x \sim Z_X}[\text{dist}(De_\theta(En_\phi(x_i)), x_i)]$ when data are drawn independently from an underlying distribution $Z_X$. The corresponding empirical loss function when we have an access to sample $\{x_i\}_{i=1}^m$ is

$$L_{\text{auto}}(\phi, \theta) := \sum_{i=1}^m \text{dist}(De_\theta(En_\phi(x_i)), x_i). \tag{1}$$

The task of finding a good autoencoder reduces to optimizing $\phi$ and $\theta$ to yield small empirical loss as in Equation 1.

We minimize Equation 1 privately using DP-SGD (described in Section 2.1). Our approach differs from previous work on private training of autoencoders (Chen et al., 2018; Acs et al., 2018; Abay et al., 2018) by *not* adding noise to the decoder during DP-SGD, whereas previous work adds noise to both the encoder and decoder. This improves performance by reducing the noise injected into the model by half, while still maintaining the same privacy guarantee (see Proposition 3). The full description of our autoencoder training is given in Algorithm 2 in the appendix. In our DP-auto-GAN framework, the autoencoder is trained first until completion, and is then fixed for the second phase of training GAN.

## 3.2 GAN TRAINING

A GAN consists of the generator $G_w$ and discriminator $D_y : \mathbb{R}^n \to \{0, 1\}$, parameterized respectively by edge weights $w$ and $y$. The aim of the generator $G_w$ is to synthesize (fake) data similar to the real dataset, while the aim of discriminator is to determine whether an input $x_i$ is from the generator's synthesized data (and assigning label $D_y(x_i) = 0$) or is real data (and assigning label $D_y(x_i) = 1$). The generator is seeded with a random noise $z \sim Z$ that contains no information about real dataset, such as a multivariate Gaussian vector, and aims to generate a distribution $G_w(z)$ that is hard for $D_y$ is distinguish from the real data. Hence, the generator wants to minimize the probability that $D_y$ makes a correct guess, $\mathbb{E}_{z \sim Z}[1 - D_y(G_w(z))]$. At the same time, the discriminator wants to maximize its probability of correct guess when the data is fake $\mathbb{E}_{z \sim Z}[1 - D_y(G_w(z))]$ and when the data is real $\mathbb{E}_{x \sim Z_X}[D_y(x)]$.

We generalize the output of $D_y$ to a continuous range $[0, 1]$, with the value indicating the confidence that a sample is real. We use the zero-sum objective for the discriminator and generator proposed by Arjovsky et al. (2017) and motivated by the Wasserstein distance of two distributions. Although their proposed Wasserstein objective cannot be computed exactly, it can be approximated by optimizing the objective:

$$\min_y \max_w O(y, w) := \mathbb{E}_{x \sim Z_X}[D_y(x)] - \mathbb{E}_{z \sim Z}[D_y(G_w(z))]. \tag{2}$$

We optimize Equation 2 privately using the DP-SGD framework described in Section 2.1. We differ from prior work on DP GANs in that our generator $G_w(\cdot)$ outputs data $G_w(z)$ in latent space $\mathbb{R}^d$ which needs to be decoded to $De(G_w(z))$ before being fed into the discriminator $D_y(z)$. The gradient $\nabla_w G_w$ is obtained by backpropagation through one more component $En(\cdot)$. Hence, the training of generator remains totally private because the additional component $En(\cdot)$ is fixed and never accesses the private data. The full description of our GAN training is given in Algorithm 4 in the appendix.

At the end of the two-phase training (including autoencoder and GAN), the noise distribution $Z$, trained generator $G_w(\cdot)$, and trained decoder $De(\cdot)$ are released to the public. The public can then generate synthetic data by sampling $z \sim Z$ to obtain a synthesized datapoint $En(G_y(z))$ repeatedly to obtain synthetic dataset of any desired size.

## 3.3 PRIVACY ACCOUNTING

Our autoencoder and GAN are trained privately by adding noise to the encoder and discriminator. Since the generator only accesses data through the discriminator's (privatized) output, then the

trained parameters of generator are also private by post-processing guarantees of differential privacy. Finally, we release the privatized decoder and generator, together with generator's noise distribution $Z$ and post-processing procedure, both of which are assumed to be public knowledge.

The privacy accounting is therefore required for the two parts that access real data $X$: training the autoencoder and the discriminator. In each training procedure, we apply the RDP accountant (described in Section 2.1 and Appendix D.2) to analyze privacy of the DP-SGD training algorithm, to compute final $(\epsilon, \delta)$-DP bound. Our application of the RDP accountant diverges from the previous literature in two main ways.

First, we do not add noise to decoder during the autoencoder training, which is contrary to prior work that adds noise to both the encoder and decoder. Our approach of not adding noise to the decoder does not affect the algorithms' overall privacy guarantees. This claim is stated formally in the following corollary, which follows immediately from Propositions 3 and 4.

**Corollary 2.** *Suppose autoencoder in DP-auto-GAN is trained with RDP privacy* $\mathrm{RDP}_{auto}(\cdot)$ *and the discriminator in DP-auto-GAN is trained with RDP privacy* $\mathrm{RDP}_D(\cdot)$, *then DP-auto-GAN is RDP with values* $\mathrm{RDP}_{auto}(\cdot)+\mathrm{RDP}_D(\cdot)$.

Second, the privacy analysis must account for two phases of training, usually with different privacy parameters (due to different batch sampling rates, noise, and number of iterations). One obvious solution is to calculate the desired $(\epsilon, \delta)$-DP parameter obtained from each phase and compose them to obtain $(\epsilon_1 + \epsilon_2, \delta_1 + \delta_2)$-DP using basic composition of differential privacy (Dwork et al., 2006). However, we can obtain a tighter privacy bound by composing the privacy at the Renyi Divergence level before translating Renyi Divergence into $(\epsilon, \delta)$-DP. In other words, we first apply Proposition 4 to compute $\mathrm{RDP}(\cdot)$ of two-phase training before applying Proposition 5 to translate RDP into DP. This is the approach highlighted in Corollary 2. In practice, this reduces the privacy parameter $\epsilon$ by about 30%.

## 4 EVALUATION METRICS

In this section, we discuss the evaluation metrics that we use in the experiments (described in Section 5) to empirically measure the quality of the synthetic data. Some of these metrics have been used in the literature, while many are novel contributions in this work. The evaluation metrics are summarized in Table 1; our contributions are in bold.

For the first two metrics described below, the dataset should be partitioned into a training set $R \in \mathbb{R}^{m_1 \times n}$ and testing set $T \in \mathbb{R}^{m_2 \times n}$, where $m = m_1 + m_2$ is the total number of samples the real data, and $n$ is the number of features in the data. After training the DP-auto-GAN, we use it to create a synthetic dataset $S \in \mathbb{R}^{m_3 \times n}$, for sufficiently large $m_3$.

**Dimension-wise probability.** This metric is used when the entire dataset is binary, and it serves as a basic sanity check to verify whether DP-auto-GAN has correctly learned the marginal distribution of each feature. Specifically, it compares the proportion of 1's (which can be thought of as estimators of Bernoulli success probability) in each feature of the training set $R$ and synthetic dataset $S$.

**Dimension-wise prediction.** This metric evaluates whether DP-auto-GAN has correctly learned the relationships *between* features. For the $k$-th feature of training set $R$ and synthetic dataset $S$, we choose $y_{R_k} \in \mathbb{R}^{m_1}$ and $y_{S_k} \in \mathbb{R}^{m_3}$ as labels of a classification or regression task based on the type of that feature, and the remaining features $R_{-k}$ and $S_{-k}$ are used for prediction. We train either a classification or regression model and measure their goodness of fit based on the model's accuracy using AUROC, $F_1$ or $R^2$ scores, which are formally defined in Appendix C.

We also propose following novel evaluation metrics. For more details, we refer the reader to Appendix D.5.

**1-way feature marginal.** This metric works as a sanity check for real features. We compute histograms for the feature interest of both real and synthetic data. The quality of the synthetic data with respect to this metric can be evaluated qualitatively through visual comparison of the histograms on real and synthetic data. This can be extended to $k$-way feature marginals and made into a quantitative measure by adding a distance measure between the histograms.

Table 1: Summary of evaluation metrics in DP synthetic data generation. We list applicability of each metric to each of the data type. Parts in **bold** are **our new contributions**. Evaluation methods with asterisk * are predictive-model-specific, and their applicability therefore depends on types of data that the chosen predictive model is appropriate for. Methods with asterisks ** are equipped with any any distributional distance of choice such as Wasserstein distance.

| TYPES | EVALUATION METHODS | DATA TYPES | | |
| --- | --- | --- | --- | --- |
| | | **Binary** | **Categorical** | **Regression** |
| Supervised | Label prediction* (Chen et al., 2018; Abay et al., 2018; Frigerio et al., 2019) | Yes | Yes | Yes |
| | Predictive model ranking* (Jordon et al., 2018) | Yes | Yes | Yes |
| Unsupervised, prediction-based | Dimension-wise prediction plot* | Yes (Choi et al. (2017), **ours**) | **Yes** | **Yes** |
| Unsupervised, distributional-distance-based | Dimension-wise probability plot (Choi et al., 2017) | Yes | No | No |
| | 3-way feature marginal, total variation distance (NIST, 2019) | Yes | Yes | Yes |
| | $k$-**way feature marginal**** | **Yes** | **Yes** | **Yes** |
| | $k$-**way PCA marginal**** | **Yes** | **Yes** | **Yes** |
| | **Distributional distance**** | **Yes** | **Yes** | **Yes** |
| **Unsupervised, qualitative** | 1-**way feature marginal (histogram)** | **Yes** | **Yes** | **Yes** |
| | 2-**way PCA marginal (data visualization)** | **Yes** | **Yes** | **Yes** |

**2-way PCA marginal.** This metric generalizes the 3-way marginal score used in NIST (2019). In particular, we compute principle components of the original data and evaluate a projection operator for first two principle components. Let us denote $P \in \mathbb{R}^{n \times 2}$ as the projection matrix such that $\overline{R} = RP$ is the projection on first two principle components of $R$. Then we evaluate projection of synthetic data $\overline{S} = SP$ and scatterplot 2-D points in $\overline{R}$ and $\overline{S}$ for visual evaluation. For quantitative evaluation, we also compute Wasserstein distance between $\overline{R}$ and $\overline{S}$. In the simulations described in Section 5, we used Wasserstein distance since we optimize for the WGAN objective, but any distributional divergence metric can be used. This approach can also be extended to $k$-way marginals by making the projection matrix $P \in \mathbb{R}^{n \times 2}$ for the first $k$ principle components.

**Distributional distance.** In this metric, we first compute the Wasserstein distance $W_2(R, S)$ between the entire real and synthetic datasets $R, S$. The Wasserstein score is then defined as

$$W_{\text{score}}(R, S) := 1 - \frac{W_2(R,S)}{\max_{x,y \in \mathcal{X}} ||x-y||_2^2},$$

where the Wasserstein distance is normalized by the maximum distance possible of two datapoints in data universe $\mathcal{X}$. To compute the Wasserstein score on $k$-way marginal PCA projection $P$, we normalize the score with additional term $\sqrt{v}$, where $v$ is the explained variance of $P$:

$$W_{\text{score}}(\bar{R}, \bar{S}, P) := 1 - \frac{W_2(\bar{R},\bar{S})}{\sqrt{v} \max_{x,y \in \mathcal{X}} ||x-y||_2^2}.$$

## 5 EXPERIMENTS

In this section we present details of our datasets and show empirical results of our experiments. Throughout our experiments, we fix $\delta = 10^{-5}$ for training DP-auto-GAN and show results for different values of $\epsilon$ including $\epsilon = \infty$ (i.e., non-private GAN) which serves as a benchmark. We also compare our results with existing works in the literature where relevant. Details of hyper-parameters and architecture can be found in the appendix. The code of our implementation is available at `https://github.com/DPautoGAN/DPautoGAN`.

## 5.1 BINARY DATA

First, we consider the MIMIC-III dataset Johnson et al. (2016) which is a publicly available dataset consisting of medical records of 46K intensive care unit (ICU) patients over 11 years old. This is a binary dataset with 1071 features. We use this dataset because it has been used in similar non-private Choi et al. (2017) and private Xie et al. (2018) GAN frameworks. We use the same evaluation metrics used in these papers. First we plot dimension-wise probability for DP-auto-GAN run on this dataset.

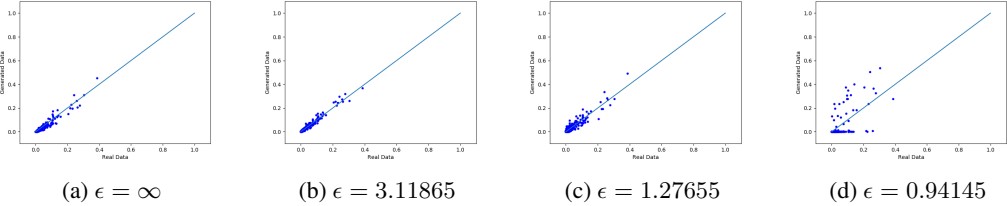

| (a) $\epsilon = \infty$ | (b) $\epsilon = 3.11865$ | (c) $\epsilon = 1.27655$ | (d) $\epsilon = 0.94145$ |

Figure 2: Dimension-wise probability scatterplots for different values of $\epsilon$. For each point in the plot represents one of the 1071 features in MIMIC-III dataset. The $x$ and $y$ coordinates of each point are the proportion of 1 in real and synthetic datasets of a feature, respectively.

As shown in Figure 2, the proportion of 1's in the marginal distribution for is similar on the real and synthetic datasets for $\epsilon = \infty$ and $\epsilon = 3.11865$, because nearly all points fall close to the line $y = x$. The performance of DP-auto-GAN is affected marginally for $\epsilon = 1.27655$ which can be noticed by increased variance of points along line $y = x$. The performance drop with $\epsilon = 0.94145$ is expected since smaller values of $\epsilon$ correspond to stronger privacy guarantees. Figure 3 shows the plots of dimension-wise prediction using DP-auto-GAN for different values of $\epsilon$. Many points are concentrated along the lower side of line $y = x$, which indicates that the AUROC score of the real dataset is close to that of the synthetic dataset. Fewer points are seen on the plots with smaller $\epsilon$ values because many features in the synthetic data have a high proportion of 0's, so the logistic regression classifier trained on these features uniformly outputs 0. In such cases, the AUROC score is $1/2$ by default and does not have any meaning, so we drop those features from the plot. We note that our results are significantly stronger than the ones obtained in Xie et al. (2018) with $\epsilon \in [96.5, 231]$ because we obtain dramatically better performance with $\epsilon$ values that are two orders of magnitude smaller. For visual performance comparison, see Figures 4 and 5 of Xie et al. (2018).

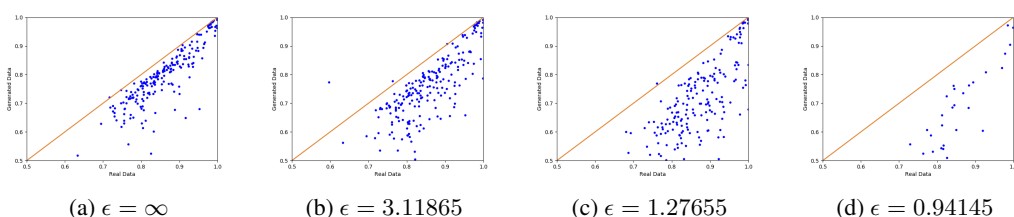

| (a) $\epsilon = \infty$ | (b) $\epsilon = 3.11865$ | (c) $\epsilon = 1.27655$ | (d) $\epsilon = 0.94145$ |

Figure 3: Dimension-wise prediction scatterplots for different values of $\epsilon$. Each point represents one of 1071 features in MIMIC-III dataset. For each point, the $x$ and $y$ coordinates represent the AUROC score of a logistic regression classifier trained on real and synthetic datasets, respectively. The line $y = x$ corresponds to the ideal performance.

## 5.2 MIXED DATA

Second, we consider the ADULT dataset Dua & Graff (2017) which is an extract of the U.S. Census and contains information about working adults. This dataset has 14 features out of which 10 features are categorical and four are real-valued. Figure 4 shows the dimension-wise prediction plot of DP-auto-GAN on this dataset. For categorical features (represented by blue points and a single green point), we use random forest classifier in order to compare our result with Frigerio et al. (2019). For real-valued features (represented by red points), we used a lasso regression model. The green point

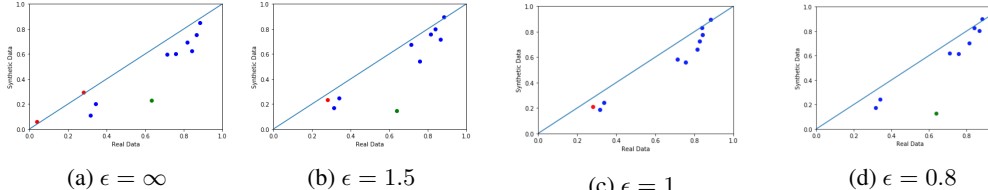

|  (a) $\epsilon = \infty$ | (b) $\epsilon = 1.5$ | (c) $\epsilon = 1$ | (d) $\epsilon = 0.8$ |

Figure 4: Dimension-wise prediction scatterplot for different values of $\epsilon$. Each point represents one of 14 features in the ADULT dataset. Blue points and single green points correspond to categorical features, and are plotted according to $F_1$ score. Red points correspond to real-valued features, and we plot $R^2$ score. For each point, $x$ and $y$ coordinate represents relevant score evaluated on real and synthetic datasets, respectively.

Table 2: Accuracy scores of prediction on salary feature evaluated on different $\epsilon$ values.

| $\epsilon$ value | Real dataset | $\infty$ | 7 | 3 | 1.5 | 1 | 0.8 |
|---|---|---|---|---|---|---|---|
| Accuracy (ours) | 86.63% | 79.18% | | | 77.86% | 76.92% | 77.7% |
| Accuracy (Frigerio et al. (2019)) | 77.2% | 76.7% | 76.0% | 75.3% | | | |

corresponds to the *salary feature* of the data, which is real-valued but treated as binary, based on the condition $> \$50k$, which is similarly used as a binary label in Frigerio et al. (2019). We use $F_1$ score as our classification accuracy measure for categorical features in in Figure 4, and we use $R^2$ score as our regression accuracy for real-valued features. The $F_1$ score is preferred over AUROC score for the ADULT dataset because it has many non-binary features where AUROC cannot be used. Each point in Figure 4 corresponds to one feature, and the $x$ and $y$ coordinates respectively show the accuracy score on the real data and the synthetic data.

Similar to the MIMIC-III dataset, we see that for large values of $\epsilon$, points are scattered close to $y = x$ line, and as $\epsilon$ gets smaller, points gradually shift downward implying, that accuracy of synthetic data deceases with stronger privacy guarantees. For the salary feature, we also compute accuracy scores for comparison with Frigerio et al. (2019). In Table 2, we report the accuracy of each synthetic dataset as well as benchmark accuracy. We see that our accuracy guarantees are higher than those of Frigerio et al. (2019) with smaller $\epsilon$ values.

Note that in the ADULT dataset, we have four real-valued features but we only plot one red point in Figure 4, corresponding to the *age feature*. The other real-valued features (capital gain, capital loss, and hours worked per week) were not included because even on the real data, we were not able to find a regression model with good fit (as measured by $R^2$ score) for these features in terms of the other features. To check whether we learned the distribution correctly for these features, we plot 1-way feature marginal histogram on each of them. See Figure 7 in Appendix C. It can be seen that DP-auto-GAN identifies the distribution in those features.

In order to understand combined performance of all features, we use two metrics. First, we show the qualitative results from 2-way PCA marginal score in Figure 5 A close qualitative inspection

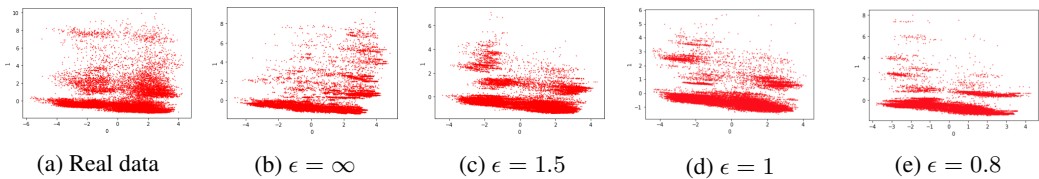

| (a) Real data | (b) $\epsilon = \infty$ | (c) $\epsilon = 1.5$ | (d) $\epsilon = 1$ | (e) $\epsilon = 0.8$ |

Figure 5: Scatterplot of projection of dataset on first two principle component of the real dataset

of plots clearly shows the similarities of trends between the plots for real dataset and for different values of $\epsilon$, as low as $\epsilon = 1$. Finally we also evaluate Wasserstein distributional distance between synthetic and real data, shown in Table 3.

Table 3: Wasserstein distance scores on 2-way PCA marginal and on whole dataset, for different $\epsilon$'s.

| Method | $\epsilon$ | 2-way PCA score | Whole-data score |
|---|---|---|---|
| DP-auto-GAN | 1.5 | 44.36% | 60.84% |
| DP-auto-GAN | 1 | 41.17% | 60.53% |
| DP-auto-GAN | 0.8 | 19.25% | 60.51% |

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

## A  ALGORITHM DESCRIPTION AND PSEUDOCODE OF DP-AUTO-GAN

We provided the pseudocode of our proposed DP-auto-GAN in Algorithm 1. The Algorithm is specified by the architecture and training parameters of encoder, decoder, generator, and discriminator. After pre-processing, DPTRAIN$_{\text{AUTO}}$ trains autoencoder fully specified in Algorithm 2. As noted earlier, the decoder is trained privately by clipping gradient norm and injecting Gaussian noise in order to obtain the gradient of decoder $g_\theta$, while the gradient of encoder $g_\phi$ can be used directly as encoder can be trained non-privately.

The second phase is to train GAN. As suggested by Goodfellow et al. (2014), discriminator trained for several iterations per one iteration of generator training. When discriminator is trained, generator is fixed, and vice-versa. Discriminator and generator training is described in Algorithms 3 and 4. As the discriminator receives real data sample in their training, the training is made private by clipping the norm and adding Gaussian noise to the gradient $g$. The training of generator does not use any private data $X$ and hence can be train without any need to clip gradient norm or to inject noise to the gradient.

Finally, the privacy analysis is via RDP accountant for each training, and composing at the RDP level (as a function of $\alpha$) as described in Corollary 2. After the sum of RDP (as a function of $\alpha$) is obtained, for any given fixed $\delta$, we optimize $\alpha$ to get the best $\epsilon$ by Proposition 5. Because the value of $\epsilon(\alpha)$ obtained from Proposition 5 as a function of $\alpha$ is convex over $\alpha$ (Van Erven & Harremos (2014) and noted by Wang et al. (2019)), we implement ternary search to efficiently optimize for $\alpha$.

**Proposition 3.** *DP-auto-GAN trained with differentially private algorithms $\mathcal{M}_1$ on the decoder and $\mathcal{M}_2$ on the discriminator (and possibly a non-private algorithm on the encoder) achieves differential privacy guarantee equivalent to that of the composition of $\mathcal{M}_1, \mathcal{M}_2$.*

*Proof.* DP-auto-GAN needs to release only generator and decoder as an output. Releasing the decoder incurs cost of privacy equal to that of $\mathcal{M}_1$. The generator accesses the data only through a discriminator, which is differentially private by mechanism $\mathcal{M}_2$, so releasing the generator has the same privacy loss as $\mathcal{M}_2$ from post-processing. Therefore, releasing both decoder and generator incurs privacy loss of composition of $\mathcal{M}_1$ and $\mathcal{M}_2$. □

Proposition 3 is stated more formally using the RDP notion of privacy (where the privacy parameters are a function of $\alpha$) in Corollary 2 in the main body. That corollary follows immediately from Propositions 3 and 4.

---

**Algorithm 1** DPAUTOGAN (full procedure)

---

1: **architecture input:** Private dataset $D \in \mathcal{X}^m$ where $\mathcal{X}$ is the set of (raw) data universe, pre-processed data dimension $n$, latent space dimension $d$, preprocessing function $Pre : \mathcal{X} \to \mathbb{R}^n$, post-processing function $Post : \mathbb{R}^n \to \mathcal{X}$, encoder architecture $En_\phi : \mathbb{R}^n \to \mathbb{R}^d$ parameterized by $\phi$, decoder architecture $De_\theta : \mathbb{R}^d \to \mathbb{R}^n$ parameterized by $\theta$, generator's noise distribution $Z$ on sample space $\Omega(Z)$, generator architecture $G_w : \Omega(Z) \to \mathbb{R}^d$ parameterized by $w$, discriminator architecture $D_y : \mathbb{R}^n \to \{0, 1\}$.
2: **autoencoder training parameters**: Learning rate $\eta_1$, number of iteration rounds (or optimization steps) $T_1$, loss function $L_{\text{auto}}$, optimization method $\text{optim}_{\text{auto}}$ batch sampling rate $q_1$ (for the batch expectation size $b_1 = q_1 m$), clipping norm $C_1$, noise multiplier $\psi_1$, microbatch size $r_1$
3: **generator training parameters**: Learning rate $\eta_2$, batch size $b_2$, loss function $L_G$, optimization method $\text{optim}_G$, number of generator iteration rounds (or optimization steps) $T_2$
4: **discriminator training parameters**: Learning rate $\eta_3$, number of discriminator iterations per generator step $t_D$, loss function $L_D$, optimization method $\text{optim}_D$, batch sampling rate $q_3$ (for the batch expectation size $b_3 = q_3 m$), clipping norm $C_3$, noise multiplier $\psi_3$, microbatch size $r_3$
5: **privacy parameter** $\delta > 0$
6: **procedure** DPAUTOGAN
7:     $X \leftarrow Pre(D)$
8:     Initialize $\phi, \theta, w, y$ for $En_\phi, De_\theta, G_w, D_y$
        ▷ *Phase 1: autoencoder training*
9:     **for** $t = 1 \ldots T_1$ **do**
10:         DPTRAIN$_{\text{AUTO}}(X, En, De, \text{autoencoder training parameters})$
        ▷ *Phase 2: GAN training*
11:     **for** $t = 1 \ldots T_2$ **do**
12:         **for** $j = 1 \ldots t_D$ **do**         ▷ *(privately) train $D_y$ for $t_D$ iterations*
13:             DPTRAIN$_{\text{DISCRIMINATOR}}(X, Z, G, De, D, \text{discriminator training parameters})$
14:         TRAIN$_{\text{GENERATOR}}(Z, G, De, D, \text{generator training parameters})$
        ▷ *Privacy accounting*
15:     $\text{RDP}_{\text{auto}}(\cdot) \leftarrow$ RDP-ACCOUNT$(T_1, q_1, \psi_1, r_1)$
16:     $\text{RDP}_D(\cdot) \leftarrow$ RDP-ACCOUNT$(T_2 \cdot t_D, q_3, \psi_3, r_3)$
17:     $\epsilon \leftarrow$ GET-EPS$(\text{RDP}_{\text{auto}}(\cdot) + \text{RDP}_D(\cdot))$
18:     **return** model $(G_w, De_\theta)$, privacy $(\epsilon, \delta)$

---

**Algorithm 2** DPTRAIN$_{\text{AUTO}}(X, En_\phi, De_\theta, \text{training parameters})$

---

1: **training parameter input**: Learning rate $\eta_1$, number of iteration rounds (or optimization steps) $T_1$, loss function $L_{\text{auto}}$, optimization method $\text{optim}_{\text{auto}}$ batch sampling rate $q_1$ (for the batch expectation size $b_1 = q_1 m$), clipping norm $C_1$, noise multiplier $\psi_1$, microbatch size $r_1$
2: **goal**: train one step of autoencoder $(En_\phi, De_\theta)$
3: **procedure** DPTRAIN$_{\text{AUTO}}$
4:     $\mathcal{B} \leftarrow$ SAMPLEBATCH$(X, q_1)$
5:     Partition $\mathcal{B}$ into $B_1, \ldots, B_k$ each of size $r$ (ignoring the dividend)
6:     $\hat{k} \leftarrow \frac{q_1 m}{r}$         ▷ an estimate of $k$
7:     **for** $j = 1 \ldots k$ **do**
        ▷ *Both $g_\phi^j, g_\theta^j$ can be computed in one backpropagation*
8:         $g_\phi^j, g_\theta^j \leftarrow \nabla_\phi(L_{\text{auto}}(De_\theta(En_\phi(B_j)), B_j)), \nabla_\theta(L_{\text{auto}}(De_\theta(En_\phi(B_j)), B_j)$
9:     $g_\phi \leftarrow \frac{1}{\hat{k}} \sum_{j=1}^k g_\phi^j$
10:     $g_\theta \leftarrow \frac{1}{\hat{k}} \left( \left( \sum_{j=1}^k \text{CLIP}(g_\phi^j, C_1) \right) + \mathcal{N}(0, C_1^2 \psi_1^2 I) \right)$
11:     $(\phi, \theta) \leftarrow \text{optim}_{\text{auto}}(\phi, \theta, g_\phi, g_\theta, \eta_1)$

---

---

**Algorithm 3** DPTRAIN$_{\text{DISCRIMINATOR}}(X, Z, G_w, De_\theta, D_y$, training parameters)

---

1: **training parameter input**: Learning rate $\eta_3$, number of discriminator iterations per generator step $t_D$, loss function $L_D$, optimization method optim$_D$, batch sampling rate $q_3$ (for the batch expectation size $b_3 = q_3 m$), clipping norm $C_3$, noise multiplier $\psi_3$, microbatch size $r_3$
2: **goal**: train one step of discriminator $D_y$
3: **procedure** DPTRAIN$_{\text{DISCRIMINATOR}}$
4:     $\mathcal{B} \leftarrow$ SAMPLEBATCH$(X, q_3)$
5:     Partition $\mathcal{B}$ into $B_1, \ldots, B_k$ each of size $r$ (ignoring the dividend)
6:     $\hat{k} \leftarrow \frac{q_1 m}{r}$                                     ▷ an estimate of $k$
7:     **for** $j = 1 \ldots k$ **do**
8:         $\{z_i\}_{i=1}^r \sim Z^r$
9:         $B' \leftarrow \{De(G_w(z_i))\}_{i=1}^r$
10:       $g^j \leftarrow \nabla_y(L_D(B_j, B', D_y))$
           ▷ In the case of WGAN,

$$L_D(B_j, B', D_y) := \frac{1}{r} \sum_{b \in B_j} D_y(b) - \frac{1}{r} \sum_{b' \in B'} D_y(b')$$

11:     $g \leftarrow \frac{1}{\hat{k}} \left( \left( \sum_{j=1}^k \text{CLIP}(g^j, C_3) \right) + \mathcal{N}(0, C_3^2 \psi_3^2 I) \right)$
12:     $y \leftarrow$ optim$_D(y, g, \eta_3)$

---

**Algorithm 4** TRAIN$_{\text{GENERATOR}}(Z, G_w, De_\theta, D_y$, generator training parameters)

---

1: **training parameter input**: Learning rate $\eta_2$, batch size $b_2$, loss function $L_G$, optimization method optim$_G$, number of generator iteration rounds (or optimization steps) $T_2$
2: **goal**: train one step of generator $G_w$
3: **procedure** TRAIN$_{\text{GENERATOR}}$
4:     $\{z_i\}_{i=1}^{b_2} \sim Z^{b_2}$
5:     $B' \leftarrow \{De(G_w(z_i))\}_{i=1}^{b_2}$
6:     $g \leftarrow \nabla_w(L_G(B', D_y))$
           ▷ In the case of WGAN,

$$L_G(B', D_y) := -\frac{1}{b_2} \sum_{b' \in B'} D_y(b')$$

7:     $w \leftarrow$ optim$_G(w, g, \eta_2)$

---

## B  MORE DETAILS ON METRIC EVALUATION

We recall the notations from the main body. We explain the scoring used more specifically.

**Dimension-wise prediction.** We describe the model's accuracy using the following well known metrics:

1. Area under the ROC curve (AUROC) score and $F_1$ score for classification: The $F_1$ score of a classifier is defined as $F_1 := \frac{2 \times \text{precision} \times \text{recall}}{\text{precision} + \text{recall}}$, where precision is ratio of true positives to true and false positives, and recall is ratio of true positives to total positives. AUROC score is graphical measure capturing area under ROC (receiver operating characteristic) curve. Both metrics take values in interval $[0, 1]$ with larger values implying good fit.

2. $R^2$ score for regression: $R^2$ score is defined as $1 - \dfrac{\sum(y_i - \widehat{y_i})^2}{\sum(y_i - \overline{y})^2}$, where $y_i$ is the true label, $\widehat{y_i}$ is the predicted label and $\overline{y}$ is the mean of the true labels. This is a popular metric used to measure goodness of fit as well as future prediction accuracy for regression.

## C  EXPERIMENTAL DETAILS

### C.1  MODEL AND TRAINING SPECIFICATION OF EXPERIMENT ON MIMIC-III DATA

The autoencoder was trained with Adam with Beta 1 = 0.9, Beta 2 = 0.999. The learning rate was set to 0.001. It was trained on minibatches of size 100 and microbatches of size 1. L2 clipping norm was set to 0.8157 and L2 penalty 0.001. The noise multiplier was changed to achieve different privacy guarantees. The layers of the autoencoder used for training were:

```
(encoder): Sequential(
(0): Linear(in-feature=60, out-feature=15, bias=True)
(1): LeakyReLU(negative-slope=0.2)
)
(decoder): Sequential(
(0): Linear(in-feature=15, out-feature=60, bias=True)
(1): Sigmoid()
)
```

The GAN was trained with optimizer RMSProp, whose Alpha value equalled 0.99. The learning rate was 0.001, training ran on minibatches of size 1,000 and microbatches of size 1. L2 clipping norm was set to 0.35, and the L2 penalty to 0.001. The L2 clipping norm was selected to be the median L2 norm observed in a non-private training loop. The noise multiplier was tuned to achieve desired privacy guarantees.

The GAN architecture is as follows:

```
Generator(
(block-0): Sequential(
(0): Linear(in-feature=64, out-feature=128, bias=False)
(1): ReLU(negative-slope=0.2)
)
(block-1): Sequential(
(0): Linear(in-feature=128, out-feature=64, bias=False)
(1): ReLU(negative-slope=0.2)
)
)

Discriminator(
(model): Sequential(
(0): Linear(in-feature=70, out-feature=256, bias=True)
(1): ReLU(negative-slope=0.2)
```

(2): Linear(in-feature=256, out-feature=1, bias=True) )
)

## C.2 ADDITIONAL MIMIC-III EMPIRICAL RESULTS

Figure 3 showed dimension-wise prediction plot for different values of $\epsilon$. As one can see, for $\epsilon = \infty$, many points are concentrated along the lower side of line $y = x$ which is the ideal performance. This shows that AUROC score of the real dataset is marginally higher than that of synthetic dataset. For $\epsilon = 3.11865$ and $\epsilon = 1.27655$, there is a gradual shift downwards compared to line $y = x$ with larger variance in the plotted points. This means that AUROC scores of real and synthetic data shows more difference for smaller values of $\epsilon$. For $\epsilon = 0.94145$, which shows the same trend, one can also see that number of datapoints plotted have reduced significantly. This is since many features in synthetic data have very high proportion of 0, so logistic regression classifier trained on these features uniformly outputs 0 on the hold-out test dataset $T$. In such cases, AUROC score outputs $1/2$ by default and as such, does not have any meaning. Hence we drop those features from the plot.

Below we show the full plots of dimension-wise prediction for MIMIC-III dataset.

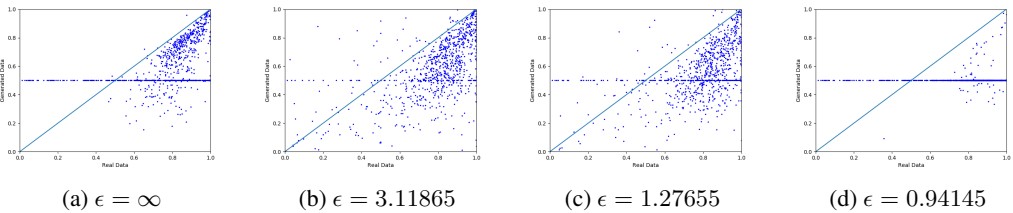

| (a) $\epsilon = \infty$ | (b) $\epsilon = 3.11865$ | (c) $\epsilon = 1.27655$ | (d) $\epsilon = 0.94145$ |

Figure 6: Full plots of dimension-wise prediction for mimic dataset

Below we provide 1-way histogram for ADULT dataset. As one can see, DP-auto-GAN identifies the marginal distribution of capital gain and capital loss quite well and it does reasonably well on hours-per-week feature.

## C.3 MODEL AND TRAINING SPECIFICATION OF EXPERIMENT ON ADULT DATA

The autoencoder was trained via Adam with Beta 1 = 0.9, Beta 2 = 0.999, and a learning rate of 0.005 for 20,000 minibatches of size 64 and a microbatch size of 1. The L2 clipping norm was selected to be the median L2 norm observed in a non-private training loop, equal to 0.012. The noise multiplier was then calibrated to achieve the desired privacy guarantee.

The GAN was composed of two neural networks, the generator and the discriminator. The generator used a ResNet architecture, adding the output of each block to the output of the following block. It was trained via RMSProp with alpha = 0.99 with a learning rate of 0.005. The discriminator was a simple feed-forward neural network with LeakyReLU hidden activation functions, also trained via RMSProp with alpha = 0.99. The L2 clipping norm of the discriminator was set to 0.022. The pair was trained on 15,000 minibatches of size 128 and a microbatch size of 1, with 15 updates to the discriminator per 1 update to the generator. Again, the noise multiplier was then calibrated to achieve the desired privacy guarantee.

A serialization of the model architectures used in the experiment can be found below.

Autoencoder(
(encoder): Sequential(
0: Linear(in-features=106, out-feature=60, bias=True)
(1): LeakyReLU(negative-slope=0.2)
(2): Linear(in-feature=60, out-feature=15, bias=True)
(3): LeakyReLU(negative-slope=0.2)
)
(decoder): Sequential(

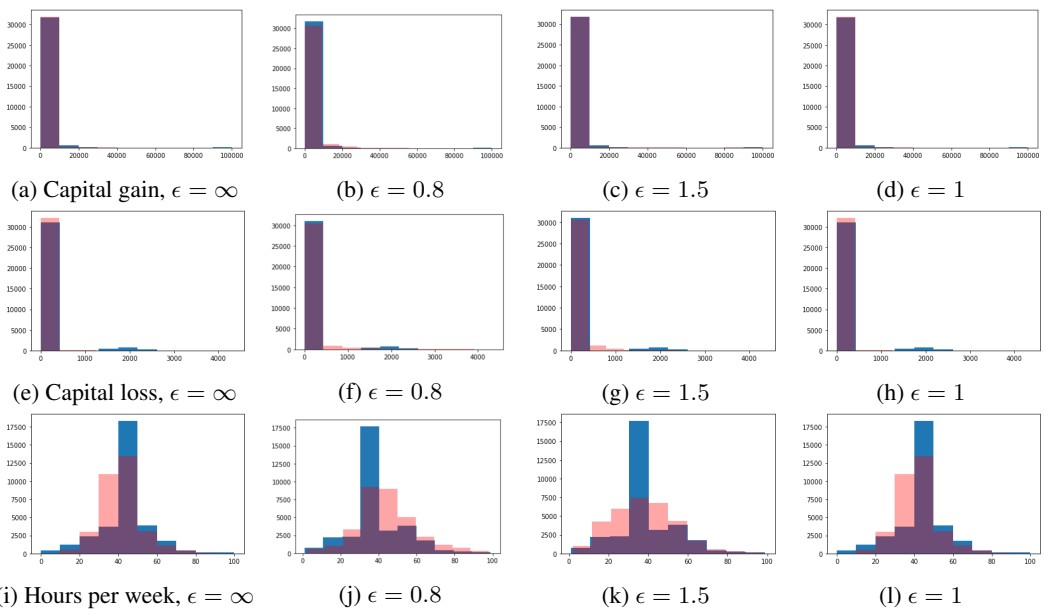

(a) Capital gain, $\epsilon = \infty$     (b) $\epsilon = 0.8$     (c) $\epsilon = 1.5$     (d) $\epsilon = 1$

(e) Capital loss, $\epsilon = \infty$     (f) $\epsilon = 0.8$     (g) $\epsilon = 1.5$     (h) $\epsilon = 1$

(i) Hours per week, $\epsilon = \infty$     (j) $\epsilon = 0.8$     (k) $\epsilon = 1.5$     (l) $\epsilon = 1$

Figure 7: 1-way histogram for different values of $\epsilon$. Three rows correspond to capital gain, capital loss and weekly work-hours

```
(0): Linear(in-feature=15, out-feature=60, bias=True)
(1): LeakyReLU(negative-slope=0.2)
(2): Linear(in-feature=60, out-feature=106, bias=True)
(3): Sigmoid()
)
)

Generator(
(block-0): Sequential(
(0): Linear(in-feature=64, out-feature=64, bias=False)
(1): BatchNorm1d()
(2): LeakyReLU(negative-slope=0.2)
)
(block-1): Sequential(
(0): Linear(in-feature=64, out-feature=64, bias=False)
(1): BatchNorm1d()
(2): LeakyReLU(negative-slope=0.2)
)
(block-2): Sequential(
(0): Linear(in-feature=64, out-feature=15, bias=False)
(1): BatchNorm1d()
(2): LeakyReLU(negative-slope=0.2)
)
)

Discriminator(
(model): Sequential(
(0): Linear(in-feature=106, out-feature=70, bias=True)
(1): LeakyReLU(negative-slope=0.2)
(2): Linear(in-feature=70, out-feature=35, bias=True)
(3): LeakyReLU(negative-slope=0.2)
(4): Linear(in-feature=35, out-feature=1, bias=True) )
)
```

# D   ADDITIONAL BACKGROUND AND RELATED WORK

## D.1   DETAILS OF DP-SGD

The DP-SGD framework (given formally in Algorithm 5) is generically applicable to private non-convex optimization.

---

**Algorithm 5** DP-SGD (one iteration step)

---

1:  **parameter input**: Data $X = \{x_i\}_{i=1}^m$, deep learning model parameter $\theta$, learning rate $\eta$, loss function $f$, optimization method OPTIM, batch sampling rate $q$ (for the batch expectation size $b = qm$), clipping norm $C$, noise multiplier $\psi$, microbatch size $r$
2:  **goal**: differentially privately train one step of the model parametrized by $\theta$ with optim
3:  **procedure** DP-SGD
4:      **procedure** SAMPLEBATCH$(X, q)$
5:          $\mathcal{B} \leftarrow \{\}$
6:          **for** $i = 1 \ldots n$ **do**
7:              Add $x_i$ to $\mathcal{B}$ with probability $q$
            **return** $\mathcal{B}$
8:      Partition $\mathcal{B}$ into $B_1, \ldots, B_k$ each of size $r$ (ignoring the dividend)
9:      $\hat{k} \leftarrow \frac{qm}{r}$  $\qquad\qquad\qquad\qquad\qquad\qquad\qquad\qquad\qquad\qquad\qquad\qquad$ ▷ an estimate of $k$
10:      $g \leftarrow \frac{1}{k} \left( \sum_{i=1}^k \text{CLIP}(\nabla_\theta f(X_{B_i}, \theta), C) + \mathcal{N}(0, C^2\psi^2 I) \right)$
11:      $\theta \leftarrow \text{OPTIM}(\theta, g, \eta)$

---

**Performance improvements.** In general, the descent step can be performed using other optimization methods—such as Adam or RMSProp—in a private manner, by replacing the gradient value with $g$ in each step. Also, one does not need to clip the individual gradients, but can instead clip the gradient of a group of datapoints, called a *microbatch* (McMahan & Andrew, 2018). Mathematically, the batch $B$ is partitioned into microbatches $B_1, \ldots, B_k$ each of size $r$, and the gradient clipping is performed on the average of each microbatch:

$$g \leftarrow \frac{1}{k} \left( \sum_{i=1}^k \text{CLIP}(\nabla_\theta f(X_{B_i}, \theta), C) + \mathcal{N}(0, C^2\psi^2 I) \right)$$

Standard DP-SGD corresponds to setting $r = 1$, but setting higher values of $r$ (while holding $|B|$ fixed) significantly decreases the runtime and reduces the accuracy, and does not impact privacy significantly for large dataset. Other clipping strategies have also been suggested. We refer readers to McMahan & Andrew (2018) for more details of clipping and other optimization strategies.

The improved privacy analysis by Abadi et al. (2016) (which has been implemented in Google (2018) and is widely used in practice) obtains a tighter privacy bound when data are subsampled, as in SGD. This analysis requires independently sampling each datapoint with a fixed probability $q$ in each step.

## D.2   CONVERTING RENYI DP TO DP

To compute the final $(\epsilon, \delta)$-differential privacy parameters from iterative runs of DP-SGD, there are three key steps.

**Step 1: Subsampled Renyi Divergence.** Given sampling rate $q$ and noise multiplier $\psi$, one can obtain RDP$(\cdot)$ values as a function of $\alpha \geq 1$ for one run of DP-SGD (Mironov, 2017). We denote this function by RDP$_{T=1}(\cdot)$, which will depend on $q$ and $\psi$.

**Step 2: Composition of RDP.** When DP-SGD is run iteratively, we can compose the Renyi privacy parameter across all runs using the following proposition.

**Proposition 4** ((Mironov, 2017)). *If $\mathcal{M}_1, \mathcal{M}_2$ respectively satisfy $(\alpha, \epsilon_1), (\alpha, \epsilon_2)$-RDP for $\alpha \geq 1$, then the composition of two mechanisms $\mathcal{M}_2(\mathcal{M}_1(X))$ satisfies $(\alpha, \epsilon_1 + \epsilon_2)$-RDP.*

Hence, we can compute RDP$(\cdot)$ values for $T$ iterations of DP-SGD as RDP-ACCOUNT$(T, q, \psi) := T \cdot \text{RDP}_{T=1}(\cdot)$.

**Step 3: Conversion to $(\epsilon, \delta)$-DP.** After obtaining the final RDP$(\cdot)$ function, any $(\alpha, \epsilon)$-RDP guarantee can be converted into $(\epsilon, \delta)$-DP.

**Proposition 5** ((Mironov, 2017)). *If $\mathcal{M}$ satisfies $(\alpha, \epsilon)$-RDP for $\alpha > 1$, then for all $\delta > 0$, $\mathcal{M}$ satisfies $(\epsilon + \frac{\log 1/\delta}{\alpha - 1}, \delta)$-DP.*

Since the $\epsilon$ privacy parameter of RDP is also a function of $\alpha$, this last step involves optimizing for the $\alpha$ that achieves smallest privacy parameter in Proposition 5.

### D.3    DIFFERENTIALLY PRIVATE GAN ARCHITECTURES

Training deep learning models reduces to minimizing some (empirical) loss function $f(X; \theta) := \frac{1}{m} \sum_{i=1}^{m} f(x_i; \theta)$ on a dataset $X = \{x_i \in \mathbb{R}^n\}_{i=1}^{m}$. Typically $f$ is a nonconvex function, and a common method to minimize $f$ is by iteratively performing stochastic gradient descent (SGD):

$$B \leftarrow \text{BATCHSAMPLE}(X) \tag{3}$$
$$\theta \leftarrow \theta - \eta \cdot \tfrac{1}{|B|} \sum_{i \in B} \nabla_\theta f(x_i, \theta) \tag{4}$$

The size of $B$ is typically fixed as a moderate number to ensure quick computation of gradient, while maintaining that $\frac{1}{|B|} \sum_{i \in B} \nabla f(x_i, \theta)$ is a good estimate of true gradient $\nabla_\theta f(X; \theta)$.

In the setting of differential privacy, $X$ is a dataset of $m$ individual's sensitive information, and two datasets are neighbors if one can be obtained from another by the addition or deletion of one datapoint. To make SGD private, a standard method proposed by Abadi et al. (2016) is to first clip the gradient of each sample to ensure the $\ell_2$-norm is at most $C$:

$$\text{CLIP}(x, C) := x \cdot \min\left(1, C/||x||_2\right).$$

Then a multivariate Gaussian noise parametrized by noise multiplier $\psi$ is added before taking an average across the batch, leading to noisy-clipped-averaged gradient estimate $g$:

$$g \leftarrow \tfrac{1}{|B|} \left( \sum_{i \in B} \text{CLIP}(\nabla_\theta f(x_i, \theta), C) + \mathcal{N}(0, C^2 \psi^2 I) \right)$$

The quantity $g$ is now private and can be used for the descent step $\theta \leftarrow \theta - \eta \cdot g$ in place of equation 4.

Variants of DP GANs have been used for synthetic data generation, including the Wasserstein GAN (WGAN) (Arjovsky et al., 2017; Gulrajani et al., 2017) and DP-WGAN (Alzantot & Srivastava, 2019; Triastcyn & Faltings, 2018) that use a Wasserstein-distance-based loss function in training (Arjovsky et al., 2017; Gulrajani et al., 2017; Alzantot & Srivastava, 2019; Triastcyn & Faltings, 2018); the conditional GAN (CGAN) (Mirza & Osindero, 2014) and DP-CGAN (Torkzadehmahani et al., 2019) that operate in a supervised (labeled) setting and use labels as auxiliary information in training; and Private Aggregation of Teacher Ensembles (PATE) (Papernot et al., 2017; 2018) for the semi-supervised setting of multi-label classification when some unlabelled public data are available (or PATEGAN (Jordon et al., 2018) when no public data are available). Our work focuses on unsupervised setting where data are unlabeled, and no (relevant) labeled public data are available.

Existing works in differentially private synthetic data generation can be summarized in Table 4.

### D.4    DIFFERENTIALLY PRIVATE TRAINING OF DEEP MODELS

There are numerous works on optimizing the performance of differentially private GANs, including data partitioning (either by class of labels in supervised setting or a private algorithm) (Yu et al., 2019; Papernot et al., 2017; 2018; Jordon et al., 2018; Abay et al., 2018; Acs et al., 2018; Chen et al., 2018); reducing the number of parameters in deep models (McMahan et al., 2017); changing the norm clipping for the gradient in DP-SGD during training (McMahan et al., 2017; van der Veen et al., 2018; Thakkar et al., 2019); changing parameters of the Gaussian noise used during training (Yu et al., 2019); and using publicly available data to pre-train the private model with a warm start (Zhang et al., 2018; McMahan et al., 2017). Clipping gradients per-layer of models (McMahan & Andrew, 2018; McMahan et al., 2017) and per-dynamic parameter grouping (Zhang et al., 2018) are also proposed. Additional details for some of these optimization approaches are given below.

Table 4: Algorithmic frameworks for differentially private synthetic data generation. **Our new algorithmic framework (in bold)** is the first to combine both DPGAN and autoencoder into one framework by using GAN to learn generative model in latent space.

| Types | Algorithmic framework | |
| --- | --- | --- |
| | **Main architecture** | **Variants** |
| Deep generative models | DPGAN (Abadi et al., 2016) | PATEGAN (Jordon et al., 2018) |
| | | DP Wasserstein GAN (Alzantot & Srivastava, 2019) |
| | | DP Conditional GAN (Torkzadehmahani et al., 2019) |
| | | Gumbel-softmax for categorical data (Frigerio et al., 2019) |
| | Autoencoder | DP-VaE (Chen et al., 2018; Acs et al., 2018) |
| | | RBM generative models in latent space (Acs et al., 2018) |
| | | Mixture of Gaussian model in latent space (Abay et al., 2018) |
| | **Autoencoder and DPGAN (ours)** | |
| Other models | SmallDB (**?**), PMW (Hardt & Rothblum, 2010), MWEM Hardt et al. (2012), DualQuery Gaboardi et al. (2014), DataSynthesizer (Ping et al., 2017), PriBayes (Zhang et al., 2017) | |

**Batch Sampling**   Three ways are known to sample a batch from data in each optimization step. The three methods are described in McMahan & Andrew (2018). We also summarize here for the completeness of DP-SGD background.

The first is to sample each individual's data with a fixed probability independently. This sampling procedure is one used in analysis of subsampled moment account in Abadi et al. (2016); McMahan & Andrew (2018) and subsampled RDP composition in Mironov (2017). This RDP composition is publicly available at Tensorflow Privacy (Google, 2018). We implement this sampling procedure and use Tensorflow Privacy to account Renyi Divergence during the training.

Another sampling policy is to sample uniformly at random a subset of fixed size of all datapoints. This achieves a different RDP guarantee from the first one, but the analysis of this sampling has been done in Wang et al. (2019).

Finally, a common subsampling procedure is to shuffle the data at random, and take a fixed-size batch in the order of the shuffling without replacement. The process is repeated after a pass over all datapoints (an epoch). Though this batch sampling is most common in practice, no subsampled privacy composition is known in this case.

**Hyperparameter Tuning**   Training a deep learning models involves hyperparameter tuning to find good architecture and optimization parameters. This process is private and privacy budget must be accounted for. Abadi et al. (2016) accounts for hyperparameter search using the work of Gupta et al. (2010). Beaulieu-Jones et al. (2019) uses Report Noisy Max Dwork & Roth (2014) to private select a model with top performance when a model evaluation metric is known. Some works are done to account for selecting high-performance models without losing much privacy (Chaudhuri & Vinterbo, 2013; Liu & Talwar, 2019). In our experimental work, we omit the privacy accounting of hyperparameter search as this is not the focus fof our contribution (new algorithmic framework using RDP subsampled composition for privacy analysis), following most literatures in differentially private synthetic data generation.

## D.5   EVALUATION METRICS FOR SYNTHETIC DATA

Now we review the evaluation schemes for measuring quality of synthetic data. Various evaluation metrics have been considered in the literature to quantify the quality of the synthetic data (Charest, 2011). Broadly, evaluation metrics can be divided into two major categories: supervised and un-

supervised. Unsupervised metrics can then be divided into three broad types: prediction-based, distributional-distance-based, and qualitative- (or visualization-) based. Metrics in previous work and our proposed metrics are summarized in Table 1.

**Supervised evaluation metrics.** These metrics are used when clear distinctions exist between feature and labels of the dataset, e.g., for healthcare applications, whether a person has a disease or not could be a label. The main aim of generating synthetic data is to best understand the relationship between features and labels. A popular metric for such cases is to train a machine learning model on the synthetic data and report its accuracy on the real test data (Xie et al., 2018). Zhang et al. (2018) used inception scores on the image data with classification tasks. Inception scores were proposed in Salimans et al. (2016) for images which measure quality as well as diversity of the generated samples. Another metric used in Jordon et al. (2018) reports whether the accuracy ranking of different machine learning models trained on the real data is preserved when the same machine learning model is trained on the synthetic data. All the evaluation metrics focus on understanding relationship between labels and features of the data and hence we call them supervised evaluation metrics. Also, in the literature, these metrics are used for classification setting but can be generalized to regression setting easily.

The disadvantage of supervised metric is that in some application, it is not clear if any feature can appropriately be a label. For example, the data analyst who wants to learn a pattern from synthetic data may not know what specific prediction tasks to perform, but rather wants to explore the data by several ways including by a unsupervised algorithm such as Principle Component Analysis (PCA). As a result, we now turn our focus to *unsupervised evaluation metric* – a metric when no feature of the data can be decisively termed as a label. We list all three types of evaluation metrics below.

**Unsupervised evaluation metric, prediction-based.** One metric of this type is proposed by Choi et al. (2017) for binary data. Instead of measuring accuracy score of one particular feature in supervised-setting, one can predict *every* single feature by using the rest of features. The prediction score is therefore created for each single feature, creating a list of dimension- (or feature-) wise prediction scores. A good synthetic data should resemble dimension-wise prediction score of that of real data. Intuitively, similar dimension-wise prediction shows that synthetic data correctly captures inter-feature relationships in the real data.

Though this was proposed for binary data, we extend this to mixed type data by allowing varieties of predictive models appropriate for each data type present in the dataset. For each feature, we try predictive models on the real dataset in order of increasing complexity until a good accuracy score is achieved. For example, to predict real-valued feature, we use linear classifier and then neural network predictor. This ensures that a choice of predictive model is first appropriate to the feature. Synthetic data is then evaluated by measuring the accuracy of the same trained predictive model, but on the synthetic data. A high accuracy score of the model on synthetic data close to original accuracy score on real data indicates that synthetic data resembles real data well.

Zhang et al. (2018) also provides a Jensen-Shannon score metric which measures the Jensen-Shannon divergence between output of a discriminating neural network on the real and synthetic dataset, and a Bernoulli random variable with $0.5$ probability. This metric differs from dimension-wise prediction in that the predictive model (discriminator) is trained over the whole dataset at once, rather than dimension-wise, to obtain a score.

**Unsupervised evaluation metric, distributional-distance-based.** Instead of computing dimension-wise prediction score, one can also compute the dimension-wise probability distribution, also proposed in Choi et al. (2017) for binary data. This metric compares the marginal distribution of real and synthetic data on each individual feature.

*3-way marginal*: Recently, NIST (2019) challenge used a 3-way marginal evaluation metric in which random three feature of the real and synthetic data $R, S$ are used to compute the total variation distance as a statistical score. This process is repeated a few times and finally, average score is returned. In particular, values of each of the three features are partitioned in 100 disjoint bins as follows:

$$B_{R,k}^i = \left\lfloor \frac{(R_k^i - R_{k,\min}) * 100}{R_{k,\max} - R_{k,\min}} \right\rfloor \text{ and } B_{S,k}^i = \left\lfloor \frac{(S_k^i - R_{k,\min}) * 100}{R_{k,\max} - R_{k,\min}}, \right\rfloor$$

where $R_k^i, S_k^i$ is the value of $i$-th datapoint's $k$-th feature in datasets $R$ and $S$. $R_{k,\min}, R_{k,\max}$ are respectively the minimum and maximum value of the $k$-th feature in $R$. For example, if $k = 1, 2, 3$ are the selected features then $i$-th data points of $R$ and $S$ are put into bins identified by a 3-tuple, $(B_{R,1}^i, B_{R,2}^i, B_{R,3}^i)$ and $(B_{S,1}^i, B_{S,2}^i, B_{S,3}^i)$, respectively.

Let $\mathcal{B}_R, \mathcal{B}_S$ be the set of all 3-tuple bins in datasets $R$ and $S$, and let $|B|$ denote number of datapoints in 3-tuple bin $B$, normalized by total number of data points. Then, the 3-way marginal metric reports the $\ell_1$-norm of the bin-wise difference of $\mathcal{B}_R$ and $\mathcal{B}_S$ as follows:

$$\sum_{B_1 \in \mathcal{B}_R} \sum_{B_2 \in \mathcal{B}_S} \mathbb{I}_{\{B_1 \in \mathcal{B}_S\}} \mathbb{I}_{\{B_2 = B_1\}} \big| |B_1| - |B_2| \big| + \sum_{B_1 \in \mathcal{B}_R} (1 - \mathbb{I}_{\{B_1 \in \mathcal{B}_S\}}) |B_1| + \sum_{B_2 \in \mathcal{B}_S} (1 - \mathbb{I}_{\{B_2 \in \mathcal{B}_R\}}) |B_2|.$$

Both aforementioned metrics involve two steps. First, a projection (or a selection of features) of data is specified, and second some statistical distance or visualization of synthetic and real data in the projected space is computed. Dimension-wise probability for binary data corresponds to projecting data into each single dimension, and visualize synthetic and real distributions in projected space by histograms (for binary data, histogram can be specified by one single number, i.e. probability of feature being 1). 3-way marginal first selects a three-dimensional space specified by three features as a space to project data to, discretize the synthetic and real distributions on that space, then compute a total variation distance between discretized distributions. Our proposed metric generalizes both steps of designing the metric as follow(s).

*Generalization of Data Projection:* One can generalize selection of 3 features (3-way marginal) to any $k$ features ($k$-way marginal). However, one can also select $k$ *principle components* instead of $k$ features. We distinguish this as $k$-way *feature* marginal (projection onto a space spanned by feature dimensions) and $k$-way *PCA* marginal (projection onto a space spanned by principle components of original datasets). Intuitively, $k$-way PCA marginal best compress information of real data in small $k$-dimension space, and hence is a better candidate for comparing projected distributions.

*Generalization of Distributional Distance:* Total variation distance can be misleading as it does not encode any information on distance of support of two distributions. In general, one can define any metric of choice (optionally with discretization) on two projected distributions, such as Wasserstein distance which also depends on distance of supports of two distributions.

Finally, we define *Distributional Distance* metric without any data projection. Computing statistical score on full-dimensional and big data is likely computationally hard. However, we can subsample uniformly at random points from two distributions to compute the score more efficiently, then average this distance over many iterations.

**Unsupervised evaluation metric, qualitative.** As mentioned earlier, dimension-wise probability is a specific application of comparing histogram under binary data. One can hence plot histogram of each feature (1-way feature marginal) for inspection. In practice, histogram visualization is particularly helpful when a feature is strongly skewed, sparse (majority is zero), and/or hard to be predicted well by predictive models. An example of this is when predictive models do not have meaningful predictive accuracy on certain features of ADULT dataset, making prediction-based metric inappropriate, but an inspection of histograms of those features on synthetic and real data indicate that synthetic data replicates those features well.

In addition, 2-way PCA marginal is a visual representation of data that explains as much variance as possible in a plane, a good trade-off between ease of visualization and information on two datasets. As mentioned earlier, a distributional distance of choice can be defined on two distributions on these two spaces to get a quantitative metric.

