# OpenReview forum: "Differentially Private Mixed-Type Data Generation For Unsupervised Learning"
_ICLR.cc/2020/Conference — Reject_

### Official Review · AnonReviewer3 · 2019-10-21
**Official Blind Review #3**

**Rating:** 1

**Review:**

The paper considers synthetic data generation using deep GANs and autoencoders that can be shared for model training.
The authors can generate mixed-type data and consider some additional metrics that allow the evaluation of the quality of synthetic data.

While the problem raised in the paper is interesting, and there some insights on what kind of metrics one should use, the article now lacks conclusions and discussion of obtained results.
In particular, there is a significant number of misprints and inconsistencies here and there (see more on this below).
Moreover, the experiments are irreproducible e.g. I could not found information about the value of reduced dimension q in the description of the experiments.
Also, there is no comparison with previous approaches (e.g. [1, 2]), only results about the proposed one are presented.

The paper will also benefit from additional rounds of proofreading:

1. The algorithm $\mathcal{M}$ is not defined. The range $Range(\mathcal{M})$ is not defined.
1. a mixture of Gaussian distribution -> a
mixture of Gaussian distributions
2. comepare -> compare, matrics -> metrics, deceases -> decreases
4. should to minimize -> should minimize
5. In the formula "(true) loss function" subscript "i" should be dropped, as we talk about $x \sim Z$, not $x_i$ here
6. Articles in many places can be improved (finding good autoencoder -> finding a good autoencoder)
7. It is possible, that in the paragraph after the formula (2) "encoder" should be replaced by "decoder".
8.  the total number of samples the real
data - > the total number of available real data samples
9. The axis labels are too small for Figure 2
10. No reference to Figure 3 in the text of the paper. For Figure 3 the most left plot has for some reason a smaller number of points. Why?
11. The selection of classifiers is not discussed. I.e. why in some cases authors use random forests (5.2), but in other logistic regression (5.1)? Also in my opinion mixing of R2 and F1 scores in one plot can be confusing.
12. No conclusion in the end


[1.] Xu et al. Modeling tabular data using conditional GAN. NeurIPS 2019
[2.] S.K.Lim et al. DOPING: Generative Data Augmentation for Unsupervised Anomaly Detection with GAN. IEEE ICDM 2019.

**Experience Assessment:**

I have published one or two papers in this area.

**Review Assessment: Checking Correctness Of Derivations And Theory:**

I carefully checked the derivations and theory.

**Review Assessment: Checking Correctness Of Experiments:**

I carefully checked the experiments.

**Review Assessment: Thoroughness In Paper Reading:**

I read the paper thoroughly.

---

### Official Review · AnonReviewer2 · 2019-10-25
**Official Blind Review #2**

**Rating:** 3

**Review:**

This paper proposed a new algorithm for synthetic data generation under differential privacy. The algorithmic architecture combines autoencoder and GAN in a way that it only needs to add the DP-SGD noise to the decoder of the autoencoder and the discriminator of the GAN. This seems to be a good idea to be explored further.

The authors claimed that the proposed new evaluation metrics are novel contributions of the paper but there is no discussion on why they are good metrics for evaluating the quality of synthetic datasets nor which metric should be used in what scenarios.

It is unclear how the experimental results (Figure 2, 3, 4 and Table 2, 3) are interpreted. The authors mentioned comparison with DP-GAN but it is not marked in the figures the performance of DP-GAN and how its results compared with DP-auto-GAN. Please clearly state what each figure means and why the results are significant.

I wonder if it is possible to have a version of GAN that also predicts the labels of data so you can use classification task as evaluation metrics, which might be easier and more interpretable.

In Section 3.1, “not adding noise to the decoder” should be encoder.
In Section 3.3, “do not add noise to decoder” should be encoder.

The presentation of the paper needs to be improved. There are too many typos and grammar mistakes. The labels of figures in the experiment section are too small. The paper does not have a conclusion section.


**Experience Assessment:**

I have read many papers in this area.

**Review Assessment: Checking Correctness Of Derivations And Theory:**

I assessed the sensibility of the derivations and theory.

**Review Assessment: Checking Correctness Of Experiments:**

I assessed the sensibility of the experiments.

**Review Assessment: Thoroughness In Paper Reading:**

I read the paper at least twice and used my best judgement in assessing the paper.

---

### Public Comment · ~Lei_Xu4 · 2019-10-15
**Comparing with existing methods**

The DP-auto-GAN model is a combination of MedGAN (Choi et al., 2017) and DP-SGD (Abadi et al. 2016). The method is neat and works well on MIMIC and ADULT datasets.

But I think the authors should compare DP-auto-GAN with multiple baselines, especially statistical methods. For example, on ADULT dataset, PrivBayes (Zhang et al., 2017) can achieve 80% accuracy with $\eps = 1.6$. Please clarify if DP-auto-GAN can outperform PrivBayes, or there are some differences in settings.

Please consider citing the following papers:
Park et al. Data synthesis based on generative adversarial networks. VLDB 2018
Xu et al. Modeling tabular data using conditional GAN. NeurIPS 2019

* In section 3.1, "by not adding noise to the decoder". I think it should be "encoder".

---

### Author Response · Authors · 2019-11-15
**Response to Reviews**

We thank the reviewers for their time and comments.  We have made a careful editing pass on the paper to make the following improvements at the reviewers' suggestion:
1. Grammatical editing -- we caught many typos including those pointed out the the reviewers
2. Comparison to existing work -- we added a more explicit comparison to other works in this area, namely Xie et al. 2018, (above Figure 3) and Frigerio et al. 2019 (Table 2). We believe this will help highlight the performance improvements achieved by our methods.
3. Fixed the typo in Section 3 where we had swapped encoder/decoder when discussing the noise addition procedure.
4. Added a discussion of Figure 3, which was previously missing
5. Added additional details in the Appendix about the implementation of our results.

---

### Decision · Program_Chairs · 2019-12-19

**Decision:**

Reject

**Comment:**

This provides a new method, called DPAutoGAN, for the problem of differentially private synthetic generation. The method uses private auto-encoder to reduce the dimension of the data, and apply private GAN on the latent space. The reviewers think that there is not sufficient justification for why this is a good approach for synthetic generation. They also think that the presentation is not ready for publication.